# The low affinity neurotrophin receptor CD271 regulates phenotype switching in melanoma

Gaetana Restivo[1], Johanna Diener[1], Phil F. Cheng[2], Gregor Kiowski[1], Mario Bonalli[1], Thomas Biedermann[3], Ernst Reichmann[2], Mitchell P. Levesque[2], Reinhard Dummer[2] & Lukas Sommer[1]

Cutaneous melanoma represents the most fatal skin cancer due to its high metastatic capacity. According to the "phenotype switching" model, the aggressive nature of melanoma cells results from their intrinsic potential to dynamically switch from a high-proliferative/low-invasive to a low-proliferative/high-invasive state. Here we identify the low affinity neurotrophin receptor CD271 as a key effector of phenotype switching in melanoma. CD271 plays a dual role in this process by decreasing proliferation, while simultaneously promoting invasiveness. Dynamic modification of CD271 expression allows tumor cells to grow at low levels of CD271, to reduce growth and invade when CD271 expression is high, and to re-expand at a distant site upon decrease of CD271 expression. Mechanistically, the cleaved intracellular domain of CD271 controls proliferation, while the interaction of CD271 with the neurotrophin receptor Trk-A modulates cell adhesiveness through dynamic regulation of a set of cholesterol synthesis genes relevant for patient survival.

[1] University of Zürich, Institute of Anatomy, Winterthurerstrasse 190, 8057 Zürich, Switzerland. [2] University of Zürich Hospital, Department of Dermatology, Gloriastrasse 31, 8091 Zürich, Switzerland. [3] University of Zürich Children's Hospital, Tissue Biology Research Unit, August Forel Strasse 7, 8008 Zürich, Switzerland. Gaetana Restivo and Johanna Diener contributed equally to this work.  Correspondence and requests for materials should be addressed to L.S. (email: lukas.sommer@anatom.uzh.ch)

In order for tumor cells to form metastases, they first have to acquire an invasive potential, which allows the cells to emigrate from the primary tumor, to reach the blood stream, and eventually to colonize distant organs, where they can build secondary tumors. In many solid cancers, acquisition of the invasive behavior is due in part to a process called epithelial-mesenchymal-transition (EMT)[1]. In melanoma a very similar phenomenon, i.e., the dynamic and reversible transition from a proliferative to an invasive state, has also been described and is known as "phenotype switching"[2–4]. As for EMT in other solid tumors, induction of ZEB, TWIST, and SNAIL transcription factor family members, as well as repression of the cell adhesion

molecule E-cadherin (CDH1), are important for melanoma progression[5]. However, in melanoma only ZEB1 and TWIST1 seem to be implicated in disease progression and metastasis, while ZEB2 expression is to the contrary lost during these processes[6]. Another crucial player in phenotype switching in melanoma is the melanocyte-specifying microphthalmia-associated transcription factor (MITF), which controls a variety of target genes involved among others, in melanocyte differentiation[7]. High expression of MITF defines a proliferative, non-invasive subpopulation of melanoma cells, whereas reduced levels of this transcription factor have been associated with increased invasiveness and metastatic behavior[8].

Melanoma cell plasticity promoted by phenotype switching also appears to underlie the frequent development of resistance to current therapies[9]. Most melanomas harbor mutations in the mitogen-activated protein kinase (MAPK) pathway, which represents the main oncogenic signaling pathway in melanoma. In particular, genetic alterations in BRAF and NRAS are most prevalent[10], and substantial efforts have been made in the clinics to develop selective inhibitors of the MAPK pathway. This has led to major advances in the treatment of patients with melanoma, resulting in improved overall survival rate[11]. Unfortunately, relapses occur in the majority of cases, indicating that some cells in the tumor mass are resistant or develop resistance to therapies[12]. Mechanistically, this has been linked to acquisition of an expression profile reminiscent of de-differentiated melanocytes[13]. In particular, high expression of MITF in melanoma cells confers high sensitivity to MAPK pathway inhibition, while MITF$^{low}$ cells are intrinsically more resistant to those treatments[13,14]. These states appear to be regulated by ZEB1: At least in some melanoma cell lines, ZEB1 overexpression induces resistance to BRAF/MEK inhibitors associated with a conversion of a MITF$^{high}$ into a MITF$^{low}$ phenotype and with high expression of the nerve growth factor receptor CD271 (also termed NGFR, p75$^{NTR}$) in resistant cells[15,16]. Likewise, recently established immunotherapies promote intrinsic changes in melanoma cells associated with tumor cell de-differentiation and resistance formation[17]. In this case, therapy-induced proinflammatory cytokines like TNFα trigger emergence of amelanotic tumors expressing high levels of CD271[9]. Establishment of resistance and overall increased CD271 expression appears to involve cellular reprogramming, as cells expressing CD271 along with other resistance markers are rare in pre-treated melanoma cell lines and patient-derived xenografts[16]. Although controversial[18], CD271 was identified before as a marker for melanoma-initiating cells, and high CD271 expression in patients was shown to correlate with increased metastasis and poor prognosis[19,20]. Intriguingly, CD271 inactivation not only resulted in decreased melanoma cell survival, but also in increased sensitivity to BRAF inhibitor treatment, suggesting that CD271 confers therapy resistance[21]. However, the function of CD271 in phenotype switching remains to be determined.

In this study, we show that CD271 is a crucial molecule in the control of melanoma cell growth vs. invasiveness. Temporal overexpression of CD271 leads to reduced proliferation and adhesion in vitro and to increased metastasis formation in vivo. Mechanistically, we found the CD271 intracellular domain (ICD) to regulate proliferation, while the interaction of CD271 with Trk-A mediates adhesion via regulating a group of cholesterol biosynthesis genes.

## Results

**CD271$^{high}$ expression is linked to an invasive signature.** To characterize CD271-expressing melanoma cells in vivo, we first determined the genetic signature of this subpopulation of cells at the time of invasion. To do so, we used an orthotopic in vivo model established in our laboratory that is able to recapitulate early events in melanoma progression, including the break out of cells from the primary tumor[22]. More precisely we seeded human melanoma cells stably expressing GFP into engineered human skin substitutes that were transplanted onto the back of immunocompromised rats (Fig. 1a). Within 6 weeks, human melanoma cells started to move from the epidermis to the dermis of the reconstituted skin, simulating an invasion process (Fig. 1b). At this time point, we dissociated the transplant and human melanoma cells were isolated based on GFP expression and separated by FACS into two populations, expressing high (CD271$^{high}$) and low levels of CD271 (CD271$^{low}$) respectively (Fig. 1c). Gene expression (Affymetrix) (Fig. 1d) and MetaCore$^{TM}$ analysis of these two populations revealed that the CD271$^{high}$ population, compared with the CD271$^{low}$ one, was enriched for processes like cell cycle, epithelial to mesenchymal transition (EMT), extracellular matrix remodeling and adhesion, and upregulation of MITF in melanoma (Fig. 1e). Figure 1f shows selected genes from MetaCore$^{TM}$ pathways with most significant differential expression. These data are consistent with a role of CD271$^{high}$ cells in proliferation and invasion.

According to the phenotype switching model[3], melanoma cells becoming invasive slow down proliferation and decrease the expression of differentiation markers. Therefore, we used TCGA data (http://cancergenome.nih.gov/) to compare CD271 expression levels with melanocytic differentiation and invasion markers (Fig. 1g–p). This analysis showed a significant inverse correlation of CD271 expression with MITF, Tyrosinase, MLANA, PMEL, and CDH1 and a positive correlation with ZEB1, TWIST1, WNT5A, c-JUN, and AXL, strengthening the association of CD271 expression with an invasive phenotype[23,24].

To assess a potential involvement of CD271 in melanoma formation, we first made use of human melanoma cell lines with different intrinsic CD271 expression levels. CD271 protein is expressed rarely in a tumor bulk[20,25] and consequently only few human melanoma cell lines derived from tumors express high levels of the protein, as revealed by Western blot analysis (Supplementary Fig. 1a). To assess the growth of cells with different CD271 levels, we injected two cell lines (CD271$^{low}$ and CD271$^{high}$, M010817 and M050829, respectively) into immunocompromised mice. To follow their growth in vivo, the cells had

---

**Fig. 1** CD271 correlates with an invasive signature and decreased proliferation. **a** Scheme illustrating the experimental approach using human organotypic skin substitutes as the basis for a humanized in vivo melanoma model to study melanoma invasion. Human melanoma cells (M070413) stably expressing GFP are mixed with human keratinocytes (HKC) and seeded onto high-density type-I collagen hydrogels containing human dermal fibroblasts (HFB). This human dermo–epidermal skin equivalent is then transplanted onto immunocompromised rats. **b** Immunofluorescent staining of GFP and CD271 on sections of transplanted human organotypic skin substitutes (M Melanoma, D Dermis). Scale bar 75 μm. **c** After excision, cells were isolated from the skin equivalent and sorted for GFP and CD271 high and low populations. **d** Gene expression analysis of the CD271 high and low populations (n = 2, Log2 ratio < −0.6, > 0.6; FDR < 0.5). **e** Pathway analysis of the CD271 high vs. low population with MetaCore$^{TM}$ (http://thomsonreuters.com/en.html). **f** Selected genes with highest significance from the MetaCore$^{TM}$ analysis representing enrichment of the EMT, Cell Adhesion, ECM Remodeling, Cell Cycle, Upregulation of MITF in melanoma and other pathways. **g–p** Correlation of RNA seq. data from TCGA for CD271 with MITF, Tyrosinase (TYR), Melan-A (MLANA), PMEL, E-cadherin (CDH1), ZEB1, TWIST1, WNT5A, c-JUN, and AXL genes

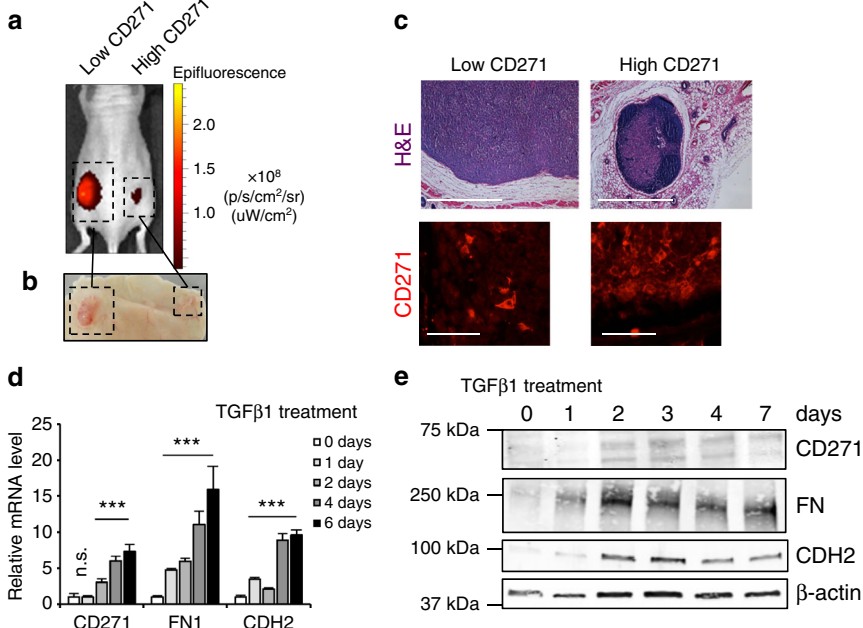

**Fig. 2** Endogenous levels of CD271 influence tumor growth. **a**, **b** Melanoma cell lines with different levels of CD271 (CD271 low = M010817; CD271 high = M050829 were stably transduced with lentiviral backbones expressing infrared fluorescent protein (iRFP) and injected in to Nude mice. The growth of the xenografts was assessed by IVIS **a** and macroscopically after euthanasia **b**. **c** H&E staining and CD271 immunofluorescence of tumors formed from low and high CD271-expressing melanoma cells. Scale bars upper panel 2 mm, scale bars lower panel 50 μm. **d**, **e** qRT-PCR and western blot of the CD271 low melanoma cell line (M010817) treated with TGFβ1 ligand (5 ng/ml) at different time points. (FN: Fibronectin; CDH2: N-cadherin; β-act.: beta actin). N = 3, P values CD271 > 0.05 and ≤0.001, P value FN ≤ 0.001, P value N-cadh. ≤0.001. Error bars indicate S.D

been stably transformed with a lentivirus carrying infrared fluorescent protein (iRFP), allowing assessment of tumor growth by an In Vivo Imaging System (IVIS)[25] (Fig. 2a). Cells expressing low levels of CD271 were fast growing and formed a 1 cm$^3$ tumor in less than four weeks. On the contrary, the cells expressing high levels of CD271 were slow growing and formed a small tumor in the same period of time (Fig. 2a, b). Histological analysis and immunohistochemical staining for CD271 confirmed the different size of tumors generated by cells expressing either low or high levels of CD271 in vivo (Fig. 2c). These results indicate that a cell line endogenously overexpressing CD271 protein proliferates slowly as compared to a cell line expressing basal levels of the protein.

Given the correlation between CD271 expression, proliferation, and invasion marker expression, we next investigated whether CD271 levels are altered in cells undergoing a transition from proliferation to invasiveness. TGFβ is known to slow down cell cycle progression and to induce an EMT-like process and invasiveness in melanoma cells[2,26]. Treatment of CD271$^{low}$ melanoma cell lines with TGFβ1 at different time points led to an increase in CD271 expression at both messenger RNA (mRNA) and protein levels in different cell lines (Fig. 2d, e and Supplementary Fig. 1b). As expected, other mesenchymal markers, including Fibronectin (FN) and N-cadherin (CDH2), were also upregulated. Similarly, an increase in CD271 protein levels was observed by knocking down the epithelial marker CDH1 either transiently by siRNA or stably by shRNA (Supplementary Fig. 1c). These results indicate that CD271 is induced when an EMT-like cascade is triggered in melanoma, in agreement with recent studies revealing high CD271 expression upon ZEB1 overexpression[15]. Taken together, our data are consistent with the idea that CD271 levels are involved in melanoma cell proliferation and invasion.

**CD271 overexpression counteracts proliferation and adhesion.** To assess whether reduced proliferation was due to an intrinsic CD271 function, we stably overexpressed this factor in a CD271$^{low}$ melanoma cell line, using a lentiviral backbone also carrying red fluorescent protein (RFP) and infrared fluorescent protein (iRFP) as markers (Fig. 3a). After 72 h of infection, CD271-overexpressing cells incorporated significantly less EdU in percentage, as compared with control cells (Fig. 3b). Moreover, when injecting CD271-overexpressing cells into immunocompromised mice, tumor growth was impaired as compared to control cells (Fig. 3c, d). Furthermore, cells overexpressing CD271 acquired a roundish morphology and eventually detached from the plate (Fig. 3e). Indeed, if transferred to adhesion-favorable fibronectin-coated plates, CD271-overexpressing cells were able to re-attach, demonstrating a CD271-induced adhesion phenotype in melanoma cells (Fig. 3f). The cells in suspension were collected and counted (Fig. 3g) and shown to represent the cell fraction with the highest CD271 expression levels, both at mRNA and protein levels (Fig. 3h, i). Although the number of cells in suspension that re-attached on fibronectin-coated plates was higher in the CD271 overexpressing cohort than the number of re-attached control cells (Fig. 3f), they were outgrown by control cells after 10 days in culture, further confirming the growth impairment of cells overexpressing CD271 (Fig. 3j). Taken together, these data show that forced expression of CD271 impairs proliferation and adhesion, consistent with an invasive phenotype of CD271$^{high}$ cells.

**Transient CD271 overexpression promotes metastasis formation.** On the basis of our findings we hypothesized that in primary tumors CD271 needs to be up-regulated in order to promote a phenotype switch from a proliferative to a less proliferative, more invasive phenotype and to be downregulated again in order to allow tumor re-growth and, hence, establishment of metastases in secondary organs. To investigate the

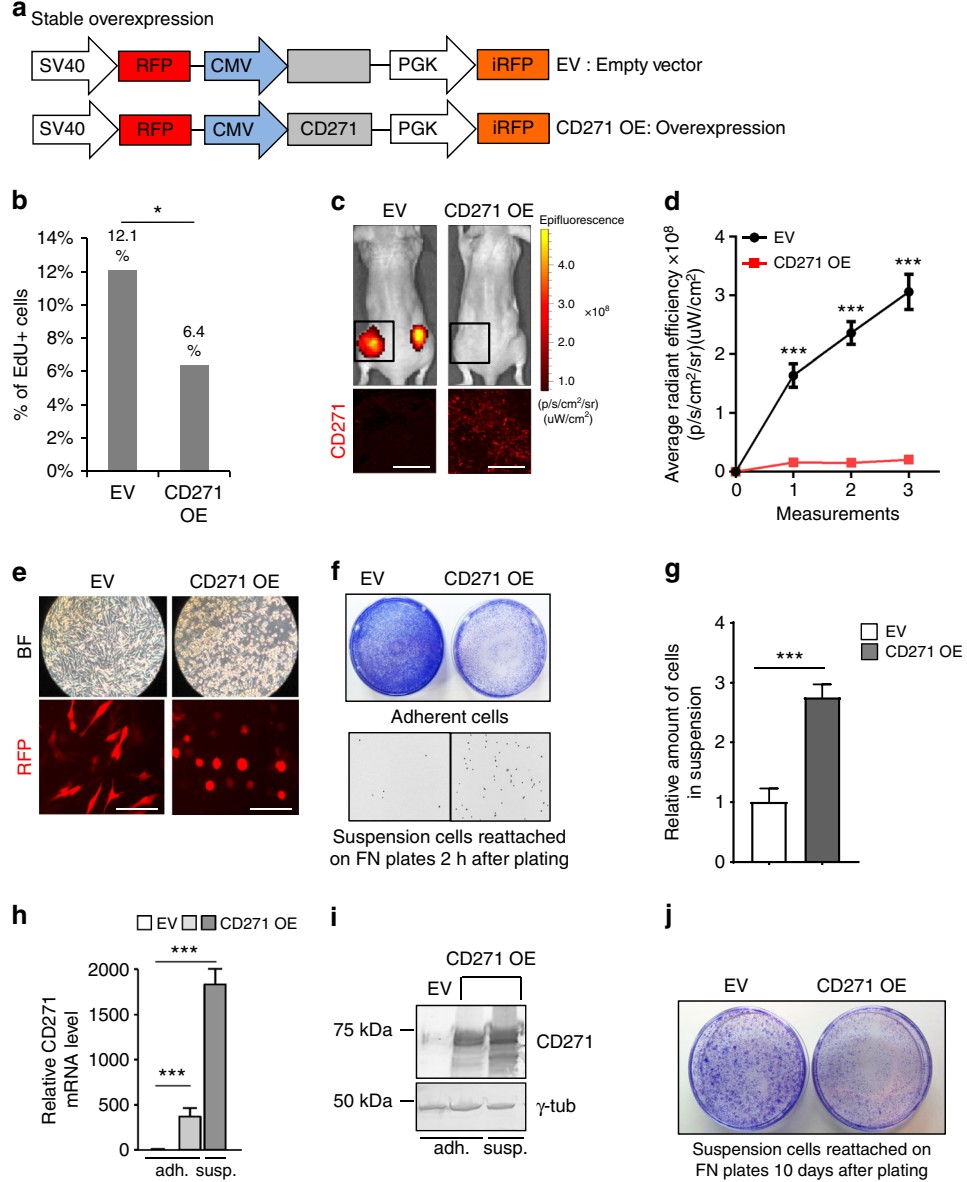

**Fig. 3** CD271 overexpression impairs proliferation and adhesion. **a** Schematic representation of the lentiviral vector used to stably overexpress CD271 in melanoma cell lines. The viral backbone carries the Red Fluorescent Protein (RFP) and infrared fluorescent protein (iRFP) reporters under the SV40 and PGK promoters respectively, as well as the CD271 overexpression cassette (CD271 OE) or an empty vector (EV) under the CMV promoter. **b** FACS analysis for EdU incorporation in cells 72 h after infection with EV or CD271 OE vectors ($n = 3$; $P$ value $\leq 0.05$). **c** Upper panel: in vivo imaging of Nude mice injected with cells carrying the empty vector (EV) or the CD271 overexpressing construct (CD271 OE). The signal is from the infrared protein present in the lentiviral backbone. Lower panel: Immunofluorescence for CD271 protein in xenografts obtained by injection of melanoma cells containing the EV or the CD271 OE vector. Scale bars 100 μm. **d** Quantification of the signal from iRFP by IVIS at different time points (four mice for a total of eight injections were analyzed for both control and CD271 overexpression; $P$ value $\leq 0.001$). Error bars indicate S.E.M. **e** Brightfield and fluorescent micrographs of melanoma cells in culture 72 h after infection with the EV or CD271 OE vector. Scale bars 50 μm. **f** Crystal violet staining of melanoma cells 72 h after infection with the EV or CD271 OE vector after washing (upper panel) and 2 h after suspension cells reattached on FN plates (lower panel visualized as inverted, black and white micrographs). **g** Quantification of the suspension cells re-attached on fibronectin-coated plates. ($n = 3$, $P$ value $\leq 0.001$) **h** qRT-PCR for CD271 levels in melanoma cells infected with EV or CD271 OE in the adherent and in the suspension fractions ($n = 3$, $P$ value $\leq 0.001$). Error bars for **g** and **h** indicate S.D. **i** Western blot for CD271 levels in melanoma cells infected with EV or CD271 OE in the adherent and in the suspension fractions. **j** Crystal violet staining of suspension cells re-attached on fibronectin-coated plates 10 days after seeding. All experiments done with cell line M010817

proposed mechanism, we engineered a TetON lentiviral expression system[25], based on the T-REx$^{\text{TM}}$ system, to transiently overexpress CD271 in melanoma cells (Fig. 4a, Supplementary Fig. 2a). With this vector system (termed CMVTOCD271), we were able to overexpress CD271 in melanoma cells by doxycycline treatment and revert its expression to basal levels by subsequently removing the compound (Supplementary Fig. 2b–d,

Supplementary Fig. 3a). Similar to the results obtained by means of stable overexpression, continuous CD271 expression in this system impaired growth in vitro and in vivo (Supplementary Fig. 2e–h, Supplementary Fig. 3b, c and h) and adhesion in vitro (Supplementary Fig. 2i, Supplementary Fig. 3d–g). To study the effect of dynamic CD271 expression on melanoma cell growth and adhesion, we transiently induced CD271 expression by short-

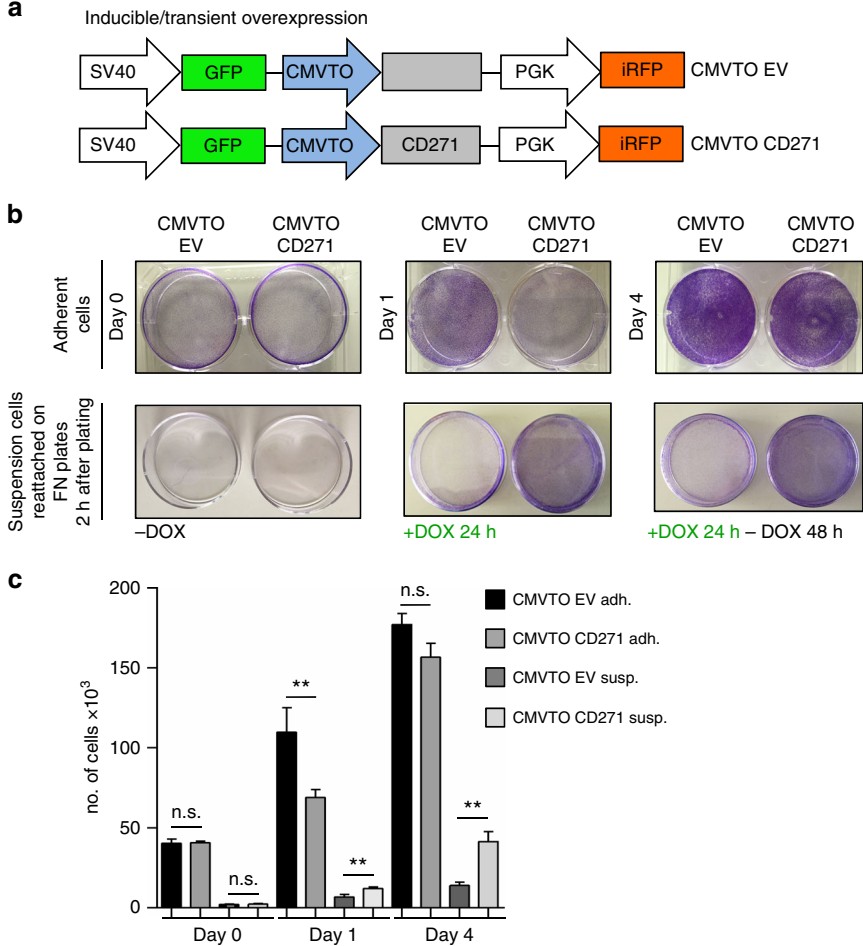

**Fig. 4** CD271 transient overexpression in vitro reveals a reversible phenotype. **a** Schematic representation of the inducible vector (TetON) to overexpress CD271 in melanoma cell lines. The viral backbone carries the green fluorescent protein (GFP) and infrared fluorescent protein (iRFP) reporters under the SV40 and PGK promoters respectively, as well as the CD271 overexpression cassette (CMVTOCD271) or an empty vector (CMVTOEV) under the inducible CMVTetOperon (CMVTO) promoter. The overexpression of the gene is controlled by doxycycline and is based on the TRex[TM] system (pLenti CMV TetR Blast, Addgene no. 17492). **b** Crystal violet staining of adherent cells (upper panels) and suspension cells forced to reattach on fibronectin-coated plates (lower panels). Not treated (day 0), treated with doxycycline for 24 h (day 1), and subsequently released from doxycycline for 48 h (day 4). **c** Quantification of cell numbers of **b** ($n = 3$, $P$ values > 0.05, ≤ 0.01). Error bars indicate S.D. All experiments done with cell line M010817

term doxycycline administration. As shown by crystal violet staining, doxycycline treatment for 1 day impaired growth of cells adherent to uncoated culture dishes (Fig. 4b) and increased the number of cells in suspension (visualized by replating on fibronectin-coated culture dishes; Fig. 4b). After removal of doxycycline, the cells regained their growth capacity, observed as a drastic increase in the number of cells both in the adherent and suspension fraction (Fig. 4b, c). Thus, sequential "off-on-off" expression of CD271 impairs proliferation and adhesion of cultured melanoma cells in a reversible manner.

Our findings suggest that transient CD271 overexpression promotes reversible phenotype switching in melanoma cells. To examine the consequences of dynamic CD271 expression in vivo, we injected cells carrying the inducible CMVTOCD271 expression construct and control cells (carrying a vector termed CMVTOEV) into immunocompromised mice. After injection, tumors were allowed to grow for 1 week before systemic doxycycline administration in drinking water for an additional week. Subsequently, the mice were released from doxycycline for 2 further weeks (Fig. 5a). As expected, the tumors initially grew in both CMVTOEV and CMVTOCD271 xenografts (Fig. 5b, c, and Supplementary Fig. 3h). After doxycycline administration, the

CMVTOCD271 tumors slowed down their growth, while the control tumors continued to grow progressively. Importantly, the CMVTOCD271 tumors started to grow again after release from doxycycline.

Our cell culture data demonstrated decreased adhesion properties of cells overexpressing CD271, which possibly might result in an increased metastatic potential in vivo. Accordingly, reversible "off-on-off" expression of CD271 would allow melanoma cells to grow at the primary injection site, to leave the primary tumor upon doxycycline-mediated CD271 over-expression, and to regain growth capacity and to form metastases at distant sites upon doxycycline release and reduction of CD271 levels. To test this hypothesis, we collected the lungs from mice that were injected either with CMVTOEV or CMVTOCD271 cells and transiently treated with doxycycline. The presence of metastatic cells in the lung was monitored by qRT-PCR for GFP (expressed from the lentiviral backbone, Fig. 4a). As shown in Fig. 5d and Supplementary Fig. 4, lungs from animals, in which CD271 had been transiently over-expressed, exhibited significantly increased GFP mRNA levels as compared with controls. Likewise, the lungs of such mice presented a high load of macroscopically visible GFP-positive

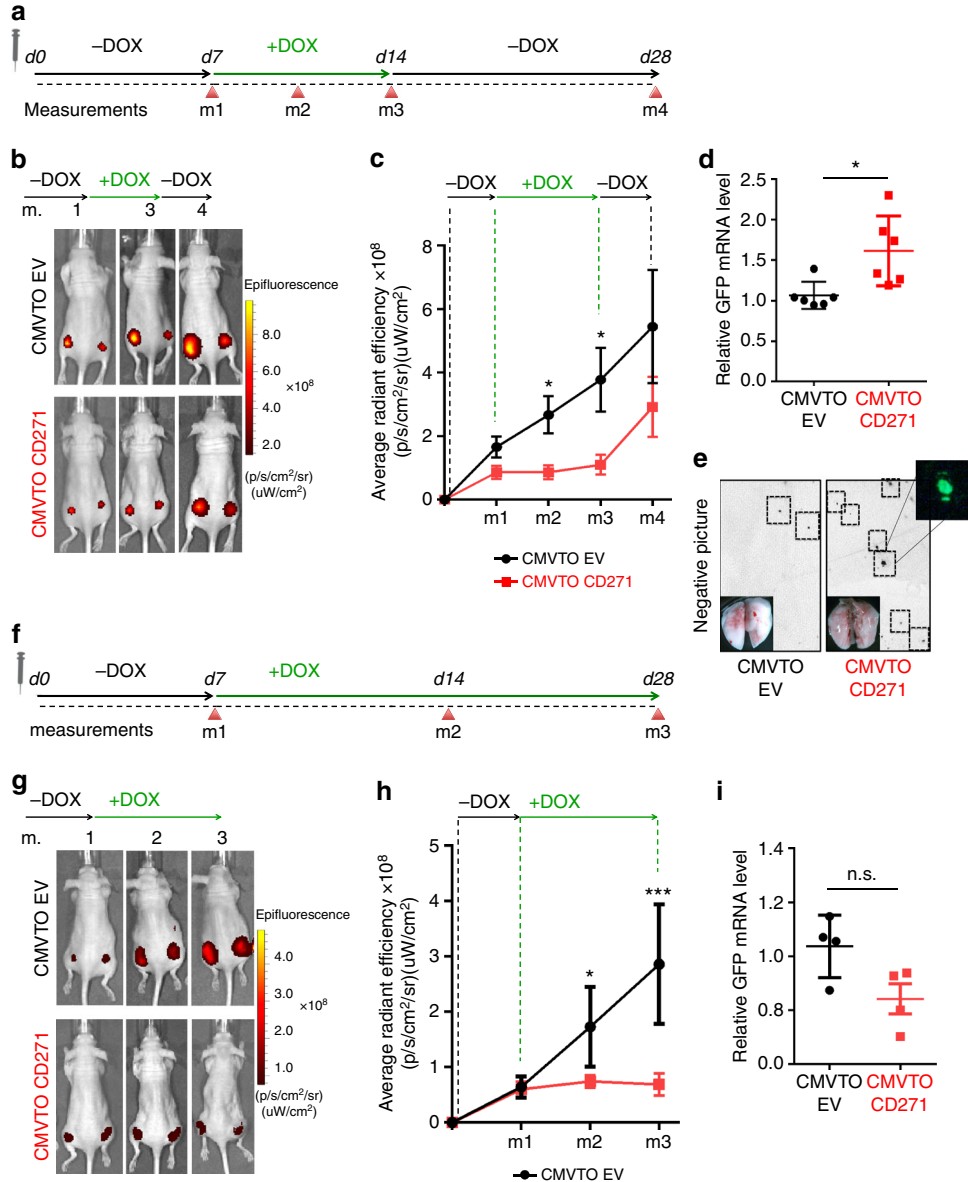

**Fig. 5** Transient CD271 overexpression in vivo increases metastasis. **a** Time course for the transient CD271 expression experiment in vivo. At day 0 (d0) the mice were injected with melanoma cells carrying CMVTOEV or CMVTOCD271 vectors. 1 week after injection (d7), the mice were treated with 200 ng/ml of doxycycline for 1 week until day 14 (d14). The mice were then released from doxycycline and analyzed 2 weeks later (d28). Different measurements (m1, m2, m3, m4) were done by IVIS to check tumor growth at different time points. **b** In vivo imaging of Nude mice injected with cells carrying CMVTOEV or CMVTOCD271 vectors. m1 = start of doxycycline treatment; m2 = during doxycycline treatment; m3 = end of doxycycline treatment; m4 = 2 weeks after release of doxycycline. **c** Quantification of the IVIS signal for infrared protein (iRFP) (4 mice for a total of 8 injections were analyzed for both conditions; $P$ value_m1 > 0.05, $P$ value_m2 ≤ 0.05, $P$ value_m3 ≤ 0.05, $P$ value_m4 > 0.05). Error bars indicate S.E.M. **d** qRT-PCR for GFP transcripts in lung lysates of Nude mice treated according to **a** ($n$ = 6 for each condition. $P$ value ≤ 0.05). Error bars indicate S.E.M. **e** Representative pictures of lungs at the end of the experiments to detect GFP. To enhance the visibility of the GFP spots, the pictures were set to grayscale and inverted. **f** Time course for the control experiment, in which CD271 expression was not switched. The experiment was designed as in **a** with the exception that between day 7 (d7) and day 28 (d28) the mice were not released from doxycycline but continuously treated. **g** In vivo imaging of Nude mice injected with cells carrying CMVTOEV or CMVTOCD271 vectors at different time points of the experiment (m1 = Start of doxycycline treatment; m2 = During doxycycline treatment; m3 = During doxycycline treatment). **h** Quantification of the IVIS signal for iRFP (4 mice for a total of 8 injections were analyzed for both conditions; $P$ value_m1 > 0.05, $P$ value_m2 ≤ 0.05, $P$ value_m3 ≤ 0.001). Error bars indicate S.E.M. **i** qRT-PCR for GFP transcripts on lung lysates of Nude mice treated according to **f**. ($n$ = 4 for each condition $P$ value > 0.05). Error bars indicate S.E.M. All experiments done with cell line M010817

metastases (Fig. 5e). In contrast, constant CD271 overexpression, starting with doxycycline administration one week after cell injection (Fig. 5f), slowed down melanoma cell proliferation in vivo (Fig. 5g, h), as expected, but did not alter the amount of GFP-expressing cells in the lungs when compared to the lungs of control mice (Fig. 5i). These data demonstrate that a switch in

CD271 expression promotes melanoma metastasis formation in distant organs.

**Different signaling axes regulate proliferation and adhesion.** CD271 is a single-pass transmembrane receptor that belongs to

the Trk receptor family. It comprises an extracellular domain able to interact with specific ligands (neurotrophins) and an intracellular domain (ICD) that is released into the cytoplasm upon cleavage by metalloproteases[27]. The proteolytic processing of CD271 involves, first, the release of a membrane-bound carbox-iterminal fragment, followed by γ-secretase-mediated cleavage, which generates a soluble ICD with signaling capabilities[28]. ICD is important for CD271-mediated control of neuronal survival and the control of neurite outgrowth[29], but is also involved in apoptotic signaling[30]. The role of ICD in cancer is still under investigation. In glioma, it has been associated with invasion[31], but its role in melanoma has not been addressed so far. To study whether ICD mediates the above-described effects of CD271 in

melanoma, we cloned this part of the CD271 receptor into a lentiviral backbone and overexpressed it in melanoma cell lines (Fig. 6a). Upon ICD overexpression cells reduced proliferation both in vitro and in vivo (Fig. 6b–d), while cellular adhesion was not affected (Fig. 6e–g). Consequently, the number of adherent cells upon ICD overexpression was lower than in the control, while the number of cells in suspension was not changed, unlike upon full-length CD271 expression.

CD271 binds neurotrophins with low affinity via its extracellular domain. By heterodimerization with other Trk receptors, its binding affinity for neurotrophins is increased. Specifically, CD271 binds NGF with Trk-A, BDNF with Trk-B and NTF-3 with Trk-C. All of these neurotrophins and their high affinity

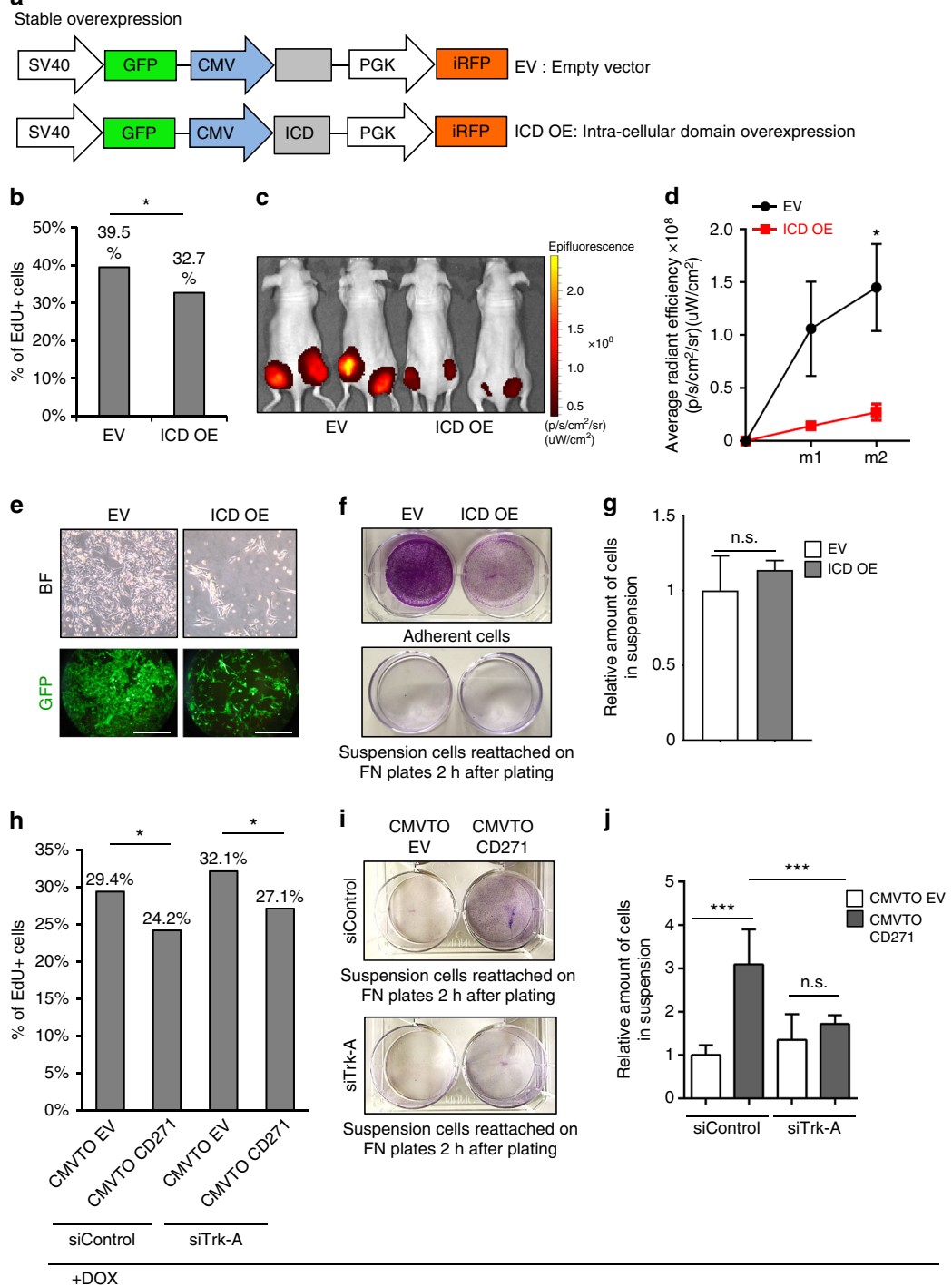

receptors are expressed by melanoma cells, suggesting autocrine signal activation[32]. It has been shown that NGF is able to induce invasion in melanoma cell lines and that this is dependent on the presence of CD271[32,33]. Furthermore, among all neurotrophins, only NGF is associated with poor prognosis when expressed at high levels in melanoma specimens, as revealed by TCGA data analysis (Supplementary Fig. 5). Therefore, we hypothesized that NGF/Trk-A in conjunction with CD271 could be involved in the cellular phenotypes seen upon CD271 overexpression. To test this idea, we overexpressed CD271 through the TetON system and at the same time knocked down Trk-A by siRNA transfection. Notably, the defect in proliferation observed upon CD271 overexpression was not rescued by Trk-A inactivation, indicating that CD271-dependent proliferation does not involve its interaction with Trk-A (Fig. 6h). In contrast, loss of adhesion induced by CD271 overexpression was significantly reverted upon simultaneous Trk-A knock down (Fig. 6i, j). Taken together, the data suggest that the two main consequences of CD271 overexpression, loss of adhesion and reduced proliferation, are mediated through two different CD271 signaling axes. While the interaction of CD271 with Trk-A is responsible for the loss of adhesion, the release of ICD is crucial for the anti-proliferative effect of CD271.

**Changes in adhesion involve cholesterol synthesis.** To address the mechanisms by which reversible "off-on-off" expression of CD271 might influence the dynamic behavior of melanoma cells, we performed RNA sequencing (RNAseq) of CMVTOCD271 cells before, during and after transient CD271 overexpression. Differentially expressed genes were clustered by considering for a first group genes significantly upregulated during transient CD271 overexpression followed by significant downregulation and, vice versa for a second set of genes, by identifying genes significantly downregulated during transient CD271 overexpression with subsequent significant upregulation (Fig. 7a). Genes with such dynamic expression patterns were termed "switching" genes and consisted of 160 genes being up- and 303 genes downregulated in a reversible manner upon short-term CD271 expression. Intriguingly, gene ontology enrichment analysis revealed that dynamic short-term CD271 overexpression regulates processes such as lipid and cholesterol biosynthesis and metabolism, rather than EMT or other well-established invasiveness programs (Fig. 7b). In particular, we found only little overlap, with inconsistent up- or downregulation, between CD271-dependent transcriptional changes and signatures previously reported to characterize proliferative vs. invasive programs in established human melanoma cell lines by Hoek et al. and Verfaillie et al.[3,34] (Supplementary Fig. 6a–d). The

transcriptional cell states defining proliferative and invasive cells, respectively, have been associated with variable expression of MITF and the receptor tyrosine kinase AXL[24]. Moreover, specific gene expression programs were identified that distinguish AXL-high/MITFlow and AXLlow/MITFhigh cells[35]. However, only 2–5% of the genes controlled by short-term CD271 overexpression represented genes of the published MITF or AXL programs (Supplementary Fig. 6e and f) and Gene Set Enrichment Analysis did not detect a significant correlation or anticorrelation between the datasets (Supplementary Fig. 6g and h). Furthermore, CD271 overexpression did not appear to influence MITF expression in tumors in vivo (Supplementary Fig. 7). Thus, short-term transient CD271 might influence processes that are either independent of previously reported invasive transcriptional programs or not represented in these programs because the latter likely reflect established states rather than transient states of cells undergoing phenotype switching.

Previous studies have reported a positive association between elevated cholesterol levels and increased cancer risk, disease progression, and metastasis formation[36,37]. Moreover, increased activity of the cholesterol pathway correlates with decreased survival of melanoma patients[38]. As shown in Fig. 7c, transient CD271 overexpression led to reversible upregulation of a set of genes crucial for cholesterol and lipid biosynthesis, many of which have been linked to cancer progression[38]. Importantly, we found that high expression of some of those key enzymes regulated by CD271 is associated with worse melanoma patient survival in TCGA (Fig. 7d). In contrast, genes annotated to cell cycle regulation were transiently downregulated during CD271-dependent phenotype switching (Fig. 7e). Of note, ICD expression in melanoma cells led to significant downregulation of most of these cell cycle regulator genes (Fig. 7f), in agreement with the anti-proliferative role of ICD described above. In contrast, ICD did not influence the expression of selected cholesterol synthesis genes (Fig. 7g). To rule out any putative effect of doxycycline on the mentioned gene sets, expression levels of CD271-controlled cell cycle regulator and cholesterol and fatty acid biosynthesis genes were measured by qRT-PCR on control cells (M010817 CMVTOEV) treated with either vehicle or doxycycline, however expression of those genes was not affected by doxycycline (Supplementary Fig. 8).

Given the suggested role of cholesterol synthesis in metastatis formation[38] and the above-mentioned CD271/Trk-A-mediated alterations in tumor cell adhesion, we next asked whether control of cholesterol synthesis by CD271 is Trk-A dependent. Strikingly, in all cases tested, the CD271-induced upregulation of cholesterol synthesis genes was reverted upon Trk-A knock down (Fig. 8a–i).

**Fig. 6** CD271 regulates proliferation and adhesion via separate signaling axes. **a** Representation of the lentiviral vector for stable overexpression of the Intra Cellular Domain (ICD) of CD271. The viral backbone contains Green Fluorescent Protein (GFP) and infrared fluorescent protein (iRFP) reporters under the SV40 and PGK promoters respectively, as well as the ICD overexpression cassette (ICD OE) or an empty vector (EV) under the CMV promoter. **b** FACS analysis for EdU incorporation in cells overexpressing ICD (ICD OE) compared to control cells (EV) ($n = 3$; $P$ value $\leq 0.05$). **c** In vivo imaging of Nude mice injected with cells carrying the empty vector (EV) or the ICD overexpressing vector (ICD OE). The signal is from the iRFP present in the lentiviral backbone. **d** Quantification of the signal obtained by IVIS for iRFP (three mice for a total of six injections were analyzed for each conditions; $P$ value_m1 $\leq 0.05$, $P$ value_m2 $\leq 0.05$). Error bars indicate S.E.M. **e** Brightfield (upper panel) and fluorescent micrographs (lower panel) of melanoma cells in culture 72 h after infection with the EV or ICD OE vector. Scale bars 100 μm. **f** Crystal violet staining of adherent cells (upper panel) and suspension cells, after re-attachment on fibronectin-coated plates (lower panel), carrying either the control vector (EV) or the ICD overexpressing vector (ICD OE). **g** Quantification of adherent and suspension cells after infection with EV or ICD OE constructs ($n = 3$, $P$ value_adh.cells $\leq 0.05$, $P$ value_susp.cells $> 0.05$). Error bars indicate S.D. **h** FACS analysis for EdU incorporation in cells infected with CMVTOEV or CMVTOCD271 constructs and transfected with sicontrol or siTrk-A. One day after siRNA transfection the cells were treated with doxycycline (1 μg/ml) for 24 h. Cells were then pulsed with EdU for 30 min and collected for FACS analysis ($n = 3$; $P$ value $\leq 0.05$). **i** Crystal violet staining of suspension cells (forced to reattach on fibronectin-coated plates) infected with CMVTOEV or CMVTOCD271 and transfected with siControl (upper panel) or siTrk-A (lower panel). The cells were treated, as described in **h**. **j** Quantification of suspension cells (CMVTOEV or CMVTOCD271 infected) after siRNA transfection and doxycycline administration (as described in **h**) ($n = 3$, $P$ value $\leq 0.05$, $P$ value $\leq 0.001$). Error bars indicate S.D. All experiments done with cell line M010817

Our data are consistent with the hypothesis, that elevated cholesterol synthesis is implicated in CD271/Trk-A-mediated changes in cell adhesion. To test this idea, we treated CMVTOEV and CMVTOCD271 cells with doxycycline and measured CD271-induced cell detachment in the presence or absence of either Lovastatin or MF-438, two established inhibitors of cholesterol synthesis[39,40]. While in the control settings (CMVTOEV), addition of cholesterol synthesis inhibitors did not elicit any overt effects on cell detachment, treatment with both Lovastatin and MF-438 rescued the loss of cell adhesion

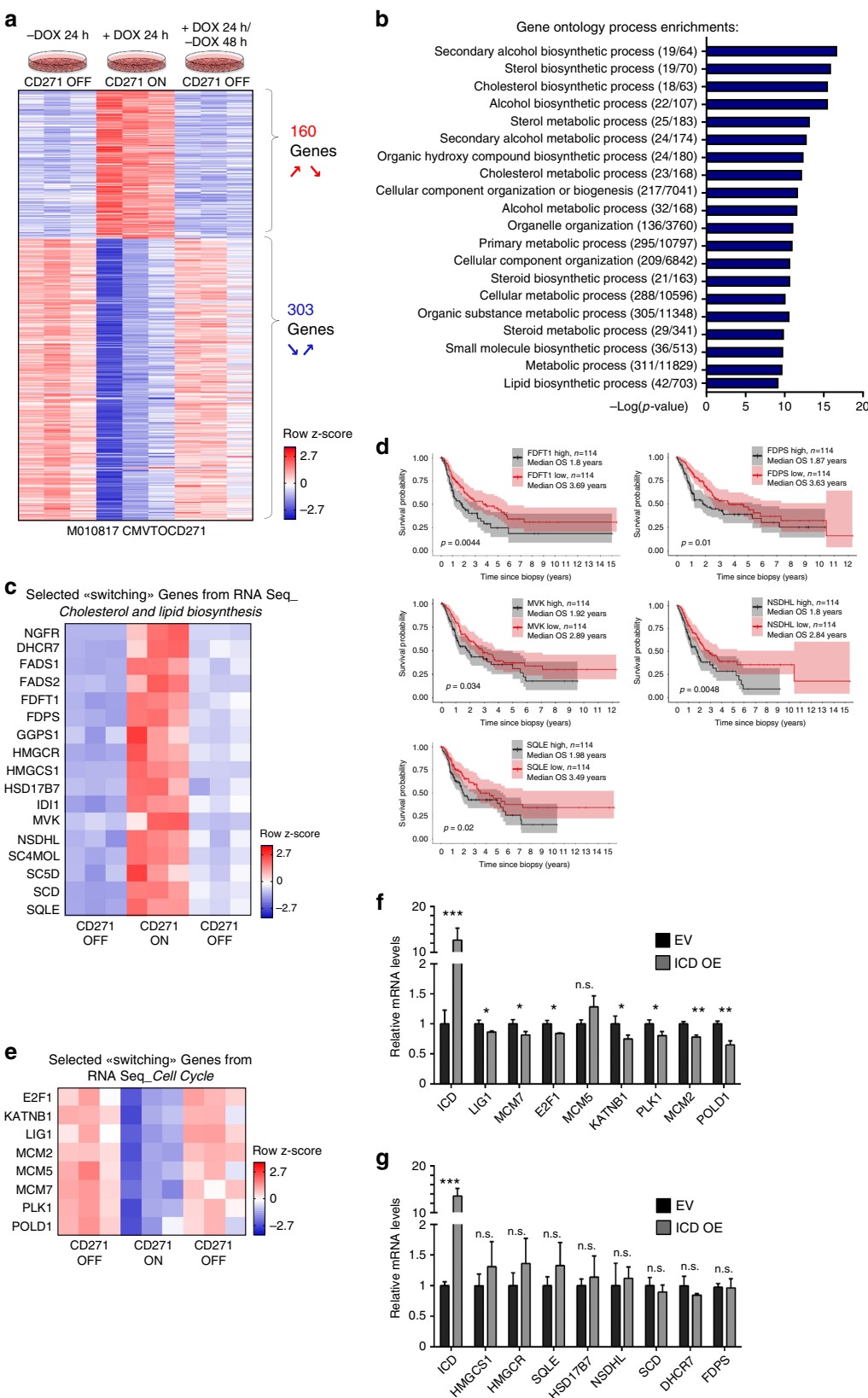

induced by CD271 overexpression (Fig. 8j, k). Thus, CD271/Trk-A appears to increase cholesterol synthesis levels, which is functionally implicated in modifying adhesion properties of melanoma cells.

## Discussion

The capacity of melanoma cells to dynamically switch between proliferative and invasive phenotypes is thought to underlie tumor progression, metastasis formation and therapy resistance in melanoma[4]. In the present study, we identify the low affinity neurotrophin receptor CD271 to play a dual role as a mediator of phenotype switching, suppressing melanoma cell proliferation while concomitantly promoting metastasis formation in vivo. CD271 is one of several molecules that have been implicated in the decision of "growing vs. going" in melanoma cells[14,41,42]. Among the best-studied pathways involved in melanoma cell plasticity is Wnt signaling, which has a canonical signaling branch acting through β-Catenin and a non-canonical branch able to suppress canonical signaling[43]. In melanoma, canonical Wnt promotes high MITF expression and tumor growth[44,45], whereas non-canonical signaling by Wnt5a and reduced MITF expression promote an invasive phenotype[23]. However, although TCGA data analysis revealed an inverse correlation between CD271 and MITF in human melanoma samples, we did not find evidence for a direct link between Wnt signaling, MITF regulation, and CD271. In particular, CD271 overexpression did not prevent MITF expression in human melanoma cells in vivo. Furthermore, reversible CD271 overexpression controlled a gene expression program hardly overlapping with previously established gene expression signatures defining proliferative vs. invasive cells[3,34,35]. Possibly, phenotype switching in melanoma involves distinct pathways able to control proliferation vs. invasiveness by independent mechanisms, although this needs to be further addressed in future work. In fact, phenotype switching in melanoma can be induced by many different stimuli, including hypoxia, therapies targeting the MAPK pathway, and immunotherapies[4,17,43,46]. Alternatively, however, the gene sets that we show to be dynamically controlled by reversible CD271 expression may characterize a state of a cell captured in the process of phenotype switching; most likely, such a state was simply not present or at least underrepresented in the long-term cell cultures from which proliferative and invasive programs were established[3,34]. This could also explain why CD271 expression itself is also not part of those published invasiveness gene expression data sets. Furthermore, cells just undergoing CD271-mediated phenotype switching are conceivably rare in the tumor bulk[16], which could account for the very limited overlap between the CD271-controlled gene expression program and the MITF and AXL programs derived from human patient biopsies[35]. In contrast, CD271 itself is part of the MITF[low]/AXL[high] program derived from single cell analysis of melanoma tissue in vivo[35], consistent with our finding that CD271 appears to be expressed in invasive cells in reconstituted skin in vivo. Possibly, the program downstream of CD271 might become attenuated with time in AXL[high] cells. Furthermore, in vivo, not all CD271/AXL[high] cells are presumably exposed to NGF or other ligands stimulating the CD271 signaling axis.

On the basis of our findings, CD271 appears to exert its two functions in phenotype switching through distinct modalities: First, heterodimerization of CD271 with the Trk-A receptor, which mediates NGF signaling, specifically affects melanoma cell invasiveness, but not proliferation. Second, ICD resulting from CD271 processing suppresses proliferation without altering adhesion and invasiveness of melanoma cells. Accordingly, inactivation of Trk-A prevented CD271 from inducing loss of adhesion, while expression of ICD in melanoma cells phenocopied the effect of CD271 on proliferation but not on invasiveness. Expression analysis of genes dynamically controlled by CD271 in an experimental set-up mimicking reversible phenotype switching revealed a set of cell cycle regulator genes that are downregulated by ICD. Although the exact mode of action of ICD in melanoma has not been investigated, the data are compatible with ICD counteracting proliferation. Likewise, genes involved in cholesterol synthesis were also dynamically controlled during phenotype switching. However, these genes turned out to be targets of Trk-A-mediated CD271 signaling and not controlled by ICD. Deregulation of cholesterol biosynthesis has been associated with several cancer types[47,48]. In melanoma, around 60% of all patients display increased expression of or copy number gains in cholesterol synthesis genes and, according to the TCGA database, high expression of a gene signature representing the activity of the cholesterol synthesis pathway correlates with decreased melanoma patient survival[38]. Cholesterol is a precursor for steroid hormones and plasma membrane components and, in addition, plays a role in intracellular signal transduction[38]. However, it remains elusive how changes in cholesterol synthesis affect tumor progression. Increases in intracellular cholesterol levels have been reported to promote tumor angiogenesis[49], survival of mesenchymal cancer cells, cancer cell migration[50], EMT[51], and metastasis formation[38,50,52]. In melanoma, high doses of statins—which efficiently block cholesterol synthesis—were reported to influence, among others, cell migration and invasion[53,54] and to reduce metastasis formation by murine B16 melanoma cells intravenously injected into syngeneic mice[55]. Likewise, an anti-tumorigenic and anti-metastatic effects of statins was also demonstrated for cancer types other than melanoma[56–58]. In agreement with these studies, we found inhibitors of cholesterol synthesis to counteract CD271/Trk-A-dependent dissemination of melanoma cells. These findings raise the intriguing question of whether changes in cholesterol and, possibly, lipid metabolism[59] are involved in melanoma phenotype switching.

**Fig. 7** Transcriptome analysis upon transient CD271 overexpression. **a** RNAseq of cells before, during and after CD271 transient overexpression was performed and analyzed using edgeR. Genes were filtered for Log2 ≥ +0.27 and ≤−0.27, p-value < 0.05, FDR < 0.05, which resulted in 160 genes significantly upregulated upon CD271 overexpression (comparing CMVTOCD271 + DOX 24 h over CMVTOCD271 –DOX24 h) with subsequent significant downregulation (comparing CMVTOCD271 + DOX 24 h/−DOX 48 h over CMVTOCD271 + DOX 24 h) and vice versa in 303 genes significantly downregulated upon CD271 overexpression with subsequent significant upregulation. Heatmap represents row z-scores of normalized counts from RNAseq. **b** Top 20 gene ontology process enrichments from MetaCore[TM]. Numbers in brackets indicate the number of differentially expressed genes above the total number of genes within an annotated pathway. **c** Selected "switching" genes from **a** involved in cholesterol and lipid biosynthesis pathways. Again, heatmap represents row z-scores of normalized counts from RNA Seq. **d** Survival curves based on TCGA data for expression of cholesterol biosynthesis genes in human melanoma samples. Data from 114 patients per condition were analyzed, which represent 25% of highest and lowest expressing patients. **e** Selected "switching" genes from **a** involved in cell cycle regulation. Again, heatmap represents row z-scores of normalized counts from RNA Seq. **f** qRT-PCR for cell cycle genes in ICD overexpressing and EV melanoma cells. **g** qRT-PCR for cholesterol and lipid biosynthesis genes in ICD overexpressing and EV melanoma cells. For **f** and **g**, n = 3 for each condition. P values > 0.05 = n.s., P values ≤ 0.05 = *, P values ≤ 0.01 = **, P values ≤ 0.001 = ***. Error bars for **f** and **g** indicate S.D. All experiments done with cell line M010817

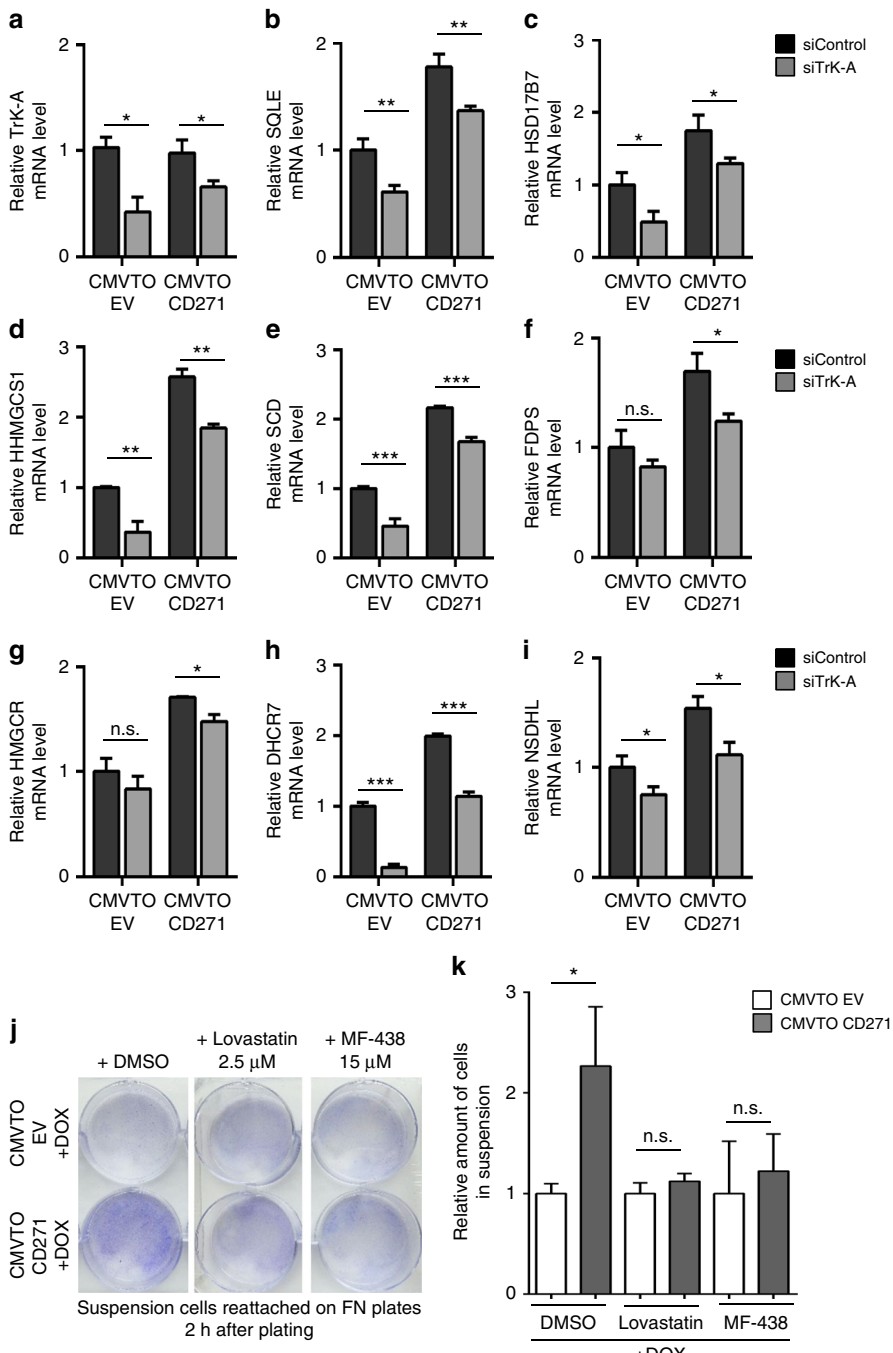

**Fig. 8** CD271-mediated upregulation of cholesterol synthesis genes is TrkA dependent and regulates cell adhesion. **a–i** qRT-PCR for cholesterol synthesis genes in CMVTOEV and CMVTOCD271 cells transfected with siCtrl or siTrkA and 24 h later treated with doxycycline (1 μg/ml) for 24 h. $N = 3$ for each condition. P values > 0.05 = n.s., P values ≤ 0.05 = *, P values ≤ 0.01 = **, P values ≤ 0.001 = ***. **j** Crystal violet staining of cells in suspension reattached on FN plates after 24 h of combinatorial treatment with doxycycline and vehicle (DMSO), Lovastatin (2.5 μM) or MF-438 (15 μM). **k** Quantification of amount of cells in adherent and suspension fractions after combinatorial cell treatment with doxycycline and vehicle (DMSO), Lovastatin or MF-438. $N = 3$ for each condition. P values > 0.05 = n.s., P values ≤ 0.05 = *. Error bars for **a–i** and **k** indicate S.D. All experiments done with cell line M010817

Our data are in line with previous studies reporting a role of NGF and/or CD271 in mediating melanoma cell invasion in vitro and of neurotrophins and their receptors in promoting melanoma cell migration[32,33,60]. However, there are also discrepancies between our work and other studies. For instance, it has been suggested based on siRNA-mediated knock down assays that signaling through Trk/CD271 receptor complexes, while promoting invasiveness, also stimulates melanoma cell proliferation[32]. Moreover, according to another study, CD271 overexpression decreased melanoma cell proliferation, as well as invasiveness[61]. Likewise, ICD has been implicated in glioma invasiveness[27], whereas we found it to affect proliferation in melanoma cells, but not invasiveness. The use of different assays for invasiveness, migration, and adhesion or of different cell lines and cell types might partially explain these discrepancies. Furthermore, the biological activity of CD271 appears to be highly dependent on its expression levels. As shown in our study, high expression of the protein leads to low proliferation and increased

invasion, while expression at basal levels defines a highly proliferative, but non-invasive state. However, we and others observed that CD271 stable knock down leading to its complete inactivation fully abolishes growth of melanoma cells[21,62]. Therefore, we advocate that a role of CD271 in phenotype switching can best be assessed by dynamically modulating its expression in vivo, as it is thought to occur in melanoma cells adopting a metastatic phenotype or in response to therapies. Indeed, constitutive expression of CD271 did not increase metastasis formation in our study, as well as in a recent report using zebrafish as model system[61]. Therefore, high CD271 levels have to revert to basal levels in metastatic cells in order for them to re-grow at distant sites and to form secondary tumors.

It has been suggested that the cellular heterogeneity of melanoma is established by melanoma cell plasticity[4,35]. EMT and metastasis formation, as well as therapy escape mechanisms presumably rely on the capacity of melanoma cells to dynamically change their phenotype in response to extracellular stimuli. This potential is reminiscent of that of normal melanocytes, which depending on the culture conditions are able to de-differentiate and to adopt features of neural crest stem cells from which melanocytes originate during embryonic development[63]. During development, neural crest stem cells become highly migratory by undergoing EMT[64] and this process is associated with upregulation of the transcription factor Sox10 and of CD271. Of note, loss of Sox10 not only impairs neural crest stem cell maintenance[65], but in melanoma also reduces the number of CD271-positive cells and counteracts tumorigenesis[66]. Furthermore, in tumors other than melanoma, the acquisition of stem cell properties has been associated with a de-differentiation process and with EMT[67]. Thus, it is conceivable that CD271-positive cells have "stem cell-like" features[20,68] although this needs to be addressed by single cell lineage-tracing experiments in vivo[69]. Importantly, our data reveal that CD271 not only marks de-differentiated melanoma cells emerging, for instance, through TGFβ-mediated EMT, BRAF inhibitor-induced reprogramming[16], or in response to immunotherapies[17], but also is functionally involved in promoting low rates of proliferation and high metastatic capacity. Therefore, the combined data suggest a potential implication of CD271 in tumor relapse after therapies. MITF induction in melanoma has been proposed as a possible therapy for melanoma because, when elevated in cancer cells, it triggers differentiation and sensitizes melanoma to TMECG, a drug inducing apoptosis[14]. As CD271 shows a role opposite to that of MITF, one possibility could be to treat melanoma with drugs inducing MITF[14] and at the same time blocking CD271 upregulation or function in order to prevent phenotype switching. Likewise, CD271 might be functionally implicated in drug-induced reprogramming of melanoma cells[16,21]. Therefore, although monotherapeutic use of statins does not seem to improve survival of melanoma patients[70], inhibition of CD271-controlled cholesterol synthesis may possibly counteract melanoma progression and metastasis formation in combination with other therapies.

## Methods

**Human organotypic skin cultures.** Organotypic cultures were prepared using a previously established transwell system consisting of 6 well cell culture inserts containing a porous membrane (3.0 μm pore-size, BD Falcon)[22]. Collagen type I hydrogels containing human dermal fibroblasts were used as dermal part of the skin substitutes. Briefly, 0.7 ml rat-tail collagen type I (3.2–3.4 mg/ml, BD Biosciences), was added to 0.2 ml chilled neutralization buffer containing 0.15 M NaOH and $1 \times 10^5$ fibroblasts. After polymerization (10 min at room temperature and 20 min at 37 °C) these dermal equivalents were grown in DMEM supplemented with 10% FCS for 5 days. Corresponding to the physiological ratio of melanocytes to keratinocytes (~1:5), $5 \times 10^4$ melanocytes or melanoma cells (M070413) were mixed with $2–2.5 \times 10^5$ keratinocytes and seeded onto each dermal equivalent. To avoid dispersion, the cells were placed into siliconized polypropylene rings of 1.5

cm diameter. After 12 h the rings were removed, 1 ml keratinocyte (Keratinocyte-SFM, Invitrogen)/melanocyte medium (Melanocyte Growth Medium, PromoCell) (ratio 5:1) was added in the upper chamber, and 2 ml was added to the lower chamber. Culturing for 2 weeks with regular medium changes gave rise to the dermo-epidermal skin substitutes used for transplantation.

**Transplantation of human organotypic skin substitutes.** The study was approved by the veterinary office of Canton of Zurich, Switzerland and was performed in accordance with Swiss law. The surgical procedure was performed, as described previously[22]. In short, a fully thickness skin wound was surgically created on the back of 10 weeks old, female athymic Nu/Nu rats (Harlan, Netherlands). To prevent wound closure from the surrounding skin, metal rings (27 mm in diameter, made to order from stainless steel at the ETH Zurich) were sutured to the skin using non-absorbable polyester sutures (Ethibond®, Ethicon). The dermo-epidermal skin substitutes were placed into the rings, fixed with 4–6 stitches, and covered with a silicone foil (Silon-SES, BMS). The rings were then covered with $5 \times 5$ cm² polyurethane sponges (Ligasano®, Ligamed) and medical strip. Sedation and anesthesia was performed as follows. Before surgery all animals were sedated with 15 mg/kg ketamine s.c. (Pfizer). Anesthesia was induced and maintained using isoflurane (Abbott), post-operative analgesia was provided by 0.05 mg/kg buprenorphine s.c. (Temgesic®, Essex). Animals were killed at the indicated time point with carbon dioxide. The grafts were excised and prepared either for immuno-histochemical processing or FACS sorting.

**Isolation of cells by FACS.** The grafts were cut in small pieces and resuspended in a solution of collagenase 3 (1 mg/ml in HBSS; ca. 10 ml per graft) for 1 h at 37 °C. Single cells from the grafts were then diluted in complete DMEM and filtered through a cell strainer (40 uM, BD falcon). They were then centrifuged and resuspended in complete medium plus EDTA (2 nM/l) then stained for CD271 (1:100; Miltenyi Biotec. APC-conjugated: 130-091-884; 30 min at 4 °C). Cells were then sorted for GFP (to specifically sort melanoma cells) and for CD271 and separated in low and high levels. For RNA extraction (for gene expression profiling) the cells were sorted in RTL buffer+beta-mercaptoethanol in the ratio 1:100 (Qiagen RNA minikit).

To sort melanoma cell lines after infection with viruses (carrying GFP or RFP), we separated the cells with PBS1X containing 2 mmol/l of EDTA. The cells were then resuspended in the same buffer and sorted then in complete RPMI (10% FCS and 1% streptomycin/penicillin plus 5 ug/ml of gentamycin). Sorting was done with FACSAria (BD Biosciences).

**Flow Cytometry for EdU detection.** Cells were pulsed with EdU (10 uM) for 30′. Cells were then treated according to the Click-iT® EdU Flow Cytometry Assay kit (C10425, C10424, ThermoScientific), measured with FACScanto II (BD Biosciences) and analyzed with Diva software (BD Biosciences).

**RNA isolation from cells and reverse transcription–qPCR.** RNA extraction and DNase treatment of samples was performed using the RNeasy Mini Kit (74104, Qiagen) and the RNase-Free DNase Set (79254, Qiagen) according to manufacturer's guidelines. Purified RNA was quantified using nanodrop and subjected to reverse transcriptase reaction using Maxima First Strand cDNA Synthesis Kit (K1641, Thermo Scientific) followed by an RNase H (EN0202, Thermo Scientific) digestion step according to manufacturer's recommendations. Real-time quantitative PCR (qRT-PCR) was performed on a LightCycler 480 System (Roche) using LightCycler 480 SYBR Green I Master (4707516001, Roche). Primers used are indicated in Supplementary Table 1. Each sample was analyzed in technical triplicates, and relative quantified RNA was normalized using β-Actin or GAPDH, as housekeeping transcripts.

**RNA isolation from lung tissue.** Lungs were collected from killed mice and immediately included in 2 ml of TRIzol® (15596026, ThermoFisher). Lungs in TRIzol® were immediately frozen at −80 °C for one night or longer and then homogenized with a tissue homogenizer (Polytron PT 2100, Kinematica). The RNA from homogenized lungs was then extracted following the manufacturer's guidelines. After extraction, 50 μg of RNA was purified and treated with DNase respectively with RNeasy Mini Kit (74104, Qiagen) and the RNase-Free DNase Set (79254, Qiagen). The reverse transcription and qRT-PCR was done, as described before.

**Microarray analysis.** Total RNA was isolated as described for reverse transcription–qPCR. Total RNA was amplified and biotin labeled using the MessageAmp II-Biotin Enhanced aRNA Amplification Kit (AM1791, Life Technologies). Biotin-labeled RNA was hybridized to Human Gene 2.1 ST Array (902136, Affymetrix) following the manufacturer's protocol. After hybridization, microarrays were washed and stained using a GeneChip Fluidics Station 450 (Affymetrix) and scanned with a GeneChip Scanner 7 G (Affymetrix). Differential gene expression was determined by R package limma. Gene ontology network analysis was performed with MetaCore™ (Thomson Reuters). Microarray data have been deposited in NCBI's Gene Expression Omnibus and are accessible through GEO

Series record number GSE103382. Supplementary Data 1 contains the list of DE genes used for MetaCore™ analysis in Fig. 1e.

**RNA sequencing**. Total RNA of three experimental replicates per condition was isolated using the RNAeasy Kit (74104, Qiagen) and RNase-Free DNase Set (79254, Qiagen) as described in the manufacturer's protocol. Quality control of total RNA was done using the Agilent RNA ScreenTape assay and the Agilent 4200 TapeStation. Enrichment of poly-A mRNA using magnetic beads (TruSeq RNA Library Prep Kit v2) was done before cDNA synthesis and library preparation. Subsequent RNA Sequencing was done with Illumina HiSeq4000 at the Functional Genomics Center Zurich, Switzerland. RNA counts were quantified from single-end reads using STAR aligner. Subsequent bioinformatic analysis of differentially expressed (DE) genes was performed with EdgeR. Gene ontology network analysis was performed with MetaCore™ (Thomson Reuters). RNA Sequencing data have been uploaded to the European Nucleotide Archive (ENA) and are accessible under the study number PRJEB22305. Supplementary Data 2 contains the list of differentially expressed genes used for MetaCore™ analysis in Fig. 7b.

**Cell cultures and reagents**. Cells used for experiments were either commercially acquired (A375) or obtained from the University Hospital Zurich where surplus tumor material was obtained after surgical removal of melanoma metastases from patients after written informed consent and approved by the local IRB (EK647 and EK800). Clinical diagnosis of the tumor material was confirmed by histology and immunohistochemistry. Primary melanoma cell cultures were established from patient biopsies using the selective adherence method (Raaijmakers et al., 2015)[71] and included in the URPP biobank, University Hospital Zürich, Department of Dermatology. Cells were grown in RPMI 1640 (Sigma Life Science, USA) supplemented with 10% fetal bovine serum (Gibco, Life Technologies, USA), 2 mM glutamine (Biochrom, Germany) and sodium pyruvate (Sigma Life Science, USA) Work with human melanoma cells was approved by the local ethical review board (KEK Nr. 2014-0425).

For experiments, all cells were cultured in growth RPMI 1640 medium supplemented with 10% FCS (16140, Life Technologies), 4 mM L-Glutamine (25030, Life Technologies) and a mix of penicillin–streptomycin antibiotics (15070, Life Technologies).

Specific information about the cell lines used in experiments can be found in Supplementary Table 2.

Cells carrying the inducible CMVTOEV and CMVTOCD271 constructs respectively, were induced in vitro with doxycycline in complete RPMI 1640 medium at a concentration of 1 µg/ml for the indicated time courses. For the detachment assay with the cholesterol inhibitors, the following final concentrations were used for 24 h treatment of the cells in combination with 1 µg/ml doxycycline: Lovastatin 2.5 µM (ab120614, Abcam), MF-438 15 µM (569406, Merck Millipore).

**Mice**. Nude mice (Hsd:AthymicNude-Foxn1nu) were purchased from Harlan and NSG (NOD.Cg-Prkdcscid Il2rgtm1Wjl/SzJ, NSG) from Charles River. Mice were housed under standard conditions with free access to water and food. Experiments were carried out with male or female mice of 6–10 weeks of age.

For the induction of CD271 in vivo mice were treated with 2 mg/ml doxycycline (D9891, Sigma-Aldrich) in drinking water plus sucrose (5%) for a maximal length of 3 weeks. All experiments with animals were approved by the veterinary office of Canton of Zurich, Switzerland and were performed in accordance with Swiss law.

**Xenografts of human melanoma cells**. Melanoma cells in culture were dissociated with PBS 2 mM EDTA in single-cell suspension. 300,000 cells were resuspended in 100 µl of RPMI-1640 medium and mix 1:1 with Matrigel matrix (356234, BD Biosciences). In total 200 µl were injected subcutaneously in both flanks of immunocompromised mice with a 1-ml syringe with a 25-gauge hypodermic needle. Tumor xenografts (1 cm$^2$) were harvested from killed mice and prepared for histological analysis.

**Histological analysis and immunofluorescence**. Xenograft samples were fixed in 4% buffered formaldehyde and embedded in paraffin. The samples were processed into sections of 4 µm thickness and slides were stained with haematoxylin and eosin (H&E) according to standard protocols. Sections were subjected to immunofluorescent analyses. Briefly, sections were deparaffinized and subjected to an antigen retrieval step using citrate buffer (S2369, Dako). Primary antibodies were applied in blocking buffer (1% BSA in PBS and 0.05% Triton X-100) overnight at 4 °C and visualized using secondary antibodies in blocking buffer for 1 h at room temperature. Subsequently, nuclei were stained with DAPI and slides were mounted with Fluorescent Mounting Medium (S3023, Dako). Sections were analyzed using a DMI 6000B microscope (Leica).

Primary antibodies are listed in Supplementary Table 3. All secondary antibodies were used at a 1:400 dilution and were purchased from Jackson ImmunoResearch Laboratories.

**MuLe cloning system**. For cloning all lentiviral vectors, we used the multiple lentiviral expression system (MuLE) that allows multiple genetic alterations to be introduced simultaneously into mammalian cells as described in Albers et al. [25].

The CD271 full length was cloned using the following primers:
Forward: EcoRI-ATGGGGGCAGGTGCCACCGG
Reverse: EcoRI- TCACACCGGGGATGTGGCAGT
The ICD of CD271 was cloned using the following primers:
Forward: EcoRI-GTGGGGCCTTGTGGCCTACATAG
Reverse: XhoI-TCACACCGGGGATGTGGCAGT

**Lentivirus production**. Lentiviruses were prepared using calcium phosphate-mediated transfection of HEK293T cells cultured in DMEM plus 10% FCS and 1% Pen/Strept antibiotics. For a 10-cm dish, lentiviral vector was co-transfected with the lentiviral packaging vector psPAX2 (12260, Addgene) and the envelope vector pMD2.G (12259, Addgene). After 24 h, fresh medium was added to the cells and 24 h later the supernatant containing the virus particles was collected and filtered through a 0.4 µM filter and either used directly to infect cells or stored at −80 °C for further infections.

**Lentiviral infection of human melanoma cells**. Human melanoma cells were grown at a confluency of 70–80% and then incubated overnight in virus-containing medium in the presence of 8 µg/ml polybrene (H9268, Sigma-Aldrich). After 24 h, fresh medium was added to the cells.

**In vivo imaging system**. Noninvasive in vivo multispectral fluorescence imaging was performed using the In vivo imaging system (IVIS) Spectrum (PerkinElmer) with Living Image software (version 4.4). Mice were anesthetized with 3% iso-flurane (Attane, MINRAD Inc.) in O$_2$ at a flow rate of 3–5 µl/min. The filter channels used for calculation of tumor growth with iRFP were Ex.675/Em.725. All measurements were performed in epifluorescence mode. For the quantification of the total radiant efficiency, a region of interest was drawn around the tumor and radiant efficiency was measured.

**Transfection of human melanoma cells with siRNA**. siRNA at a final concentration of 50 nM was used to transfect cells at 50% confluency. For transfection we used the JetPrime transfection kit (114, Polyplus transfection) and we followed the manufacturer's guidelines. siCD271, siCDH1, and siTrk-A were all purchased from Life Technology (Supplementary Table 4).

**Protein isolation and western blotting**. Cells were lysed in RIPA buffer (89900, Thermo Scientific) containing Halt Phosphatase and Protease Inhibitor Cocktail (78420, 87786, Thermo Scientific). Primary antibodies were applied in Odyssey blocking buffer (927–40000, LI-COR Biosciences) overnight at 4 °C and visualized using secondary antibodies in Odyssey blocking buffer for 45 min at room temperature. Blots were scanned and quantified with an Odyssey imaging system (LI-COR Biosciences). Quantified band intensities were normalized using either β-Actin or γ-Tubulin as housekeeping proteins. The antibodies used for western blotting are listed in Supplementary Table 5. Original western blots used in different figures can be seen in Supplementary Figure 9.

**Statistical analysis**. All statistical evaluations were carried out using GraphPad Prism 5.0 and Excel. All analyses are unpaired, two-tailed Student's t-tests. $P$ values ≤ 0.05 were considered significant (see below). All experiments with cell lines were done in triplicates and error bars represent the mean ± S.D. For the in vivo imaging at least 4 injections per group were analyzed and error bars represent the mean ± S.E.M. For the lung analysis at least three animals were used and the error bars represent the mean ± SEM. $P$ values > 0.05 = n.s., $P$ values ≤ 0.05 = *, $P$ values ≤ 0.01 = **, $P$ values ≤ 0.001 = ***.

**Data availability**. The data sets generated during the current study are available in the Gene Expression Omnibus repository (Microarray Fig. 1) under accession number GSE103382 and in the European Nucleotide Archive (RNA Sequencing Fig. 7) under project accession number PRJEB22305. All other remaining data are available within the Article and Supplementary Files, or available from the authors upon request.

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

## Acknowledgements

We thank Prof. Ian Frew and Dr Joachim Albers for helping with cloning and providing the plasmids for the MuLE system; Dr Daniel Zingg and Dr Luis Zurkirchen for inputs and discussion of the results; Jessica Häusel and Marcel Balz for assistance with histology and technical support; the Center for Microscopy and Image Analysis (ZMB) for help with slide scanning and the Functional Genomics Center Zurich (FGCZ) for assistance with the microarray and RNA Sequencing analysis. This work was supported by the Swiss National Science Foundation (SNF), partially through the Marie Heim-Vögtlin (MHV) grant; the University Research Priority Program (URPP) "Translational Cancer Research" and the Swiss Cancer League.

## Author contributions

G.R., J.D. and L.S. designed the experiments; G.R., J.D., G.K. and T.B. performed the experiments; G.R., J.D., G.K., M.B., T.B., P.F.C. and L.S. analyzed the data. M.P.L. and R. D. provided the human melanoma samples. T.B. and E.R. provided expertise in transplantation of skin substitute in rats. G.R., J.D. and L.S. wrote the manuscript.
