## [Peer Review file · Nature Communications]

Reviewers' comments:

Reviewer #1 (Remarks to the Author):

In their manuscript entitled "The low affinity neurotrophin receptor CD271 regulates phenotype switching in melanoma", Lukas Sommer and co-workers investigated the role of the low affinity neurotrophin receptor CD271 in melanoma pathogenesis.

Using an orthotopic in vivo model of human melanoma that recapitulates the first steps of melanoma metastasis the authors show that CD271+ melanoma cells express markers indicative of an invasive phenotype. Transient reversible overexpression of CD271 was able to dynamically modulate the invasive melanoma cell phenotype and promote the development of metastasis. Mechanistic analyses demonstrated that cleavage of the CD271 intracellular domain was crucial for the anti-proliferative effect of CD271 while the interaction of CD271 with Trk-A is responsible for the loss of adhesion.

Overall, the work is methodically of very high quality and addresses a clinically highly relevant topic which provides novel insights into the mechanisms of melanoma cell phenotype switching in the process of metastatic progression.

Reviewer #2 (Remarks to the Author):

In this manuscript, Restivo et al. explore the role of CD271 as a putative modulator of phenotype switching in melanoma. They provide evidence that CD271 plays a dual role in this process by mitigating cell proliferation on the one hand and stimulating invasion on the other.

This is certainly a nice piece of work, which has important clinical relevance. There are a number of concerns that would need to be addressed.

-Throughout the paper the authors refer to melanoma "phenotype switching" and study the involvement of CD271 in this process. It is claimed that CD271 can promote the proliferative to invasive switch. However, I am concerned that expression of CD271 is absent from the invasive expression datasets published by others including Hoek and colleagues themselves (45 gene sized INV list) and more recently Verfaillie et al. According to these data CD271 is not a classical invasive gene. Are cells expressing CD271 in a true invasive cell state as previously defined or an intermediary state – or different (de-differentiation) state that has not been extensively characterized previously? I suggest that the authors establish the transcriptome (and if possible a H3K27ac profile) of melanoma cells expressed CD271 ectopically and compare the gene expression/chromatin signature they obtain with the Hoek and Verfaillie datasets. Or even better, using the TetO system show that the transcriptome of melanoma cells can switch back and forth between the invasive and proliferative cell states by comparing profiles (GSEA) with the ones described by Hoek and Verfaillie.

-TGF β was used to induce phenotype switching. However, this is not what most people have been using in the field. It is suggested that CD271 expression is analyzed in cells exposed to TNF α , Hypoxia and BRAF/MEK-inhibitors.

The latter experiment may allow the authors to study the potential role of CD271 in therapy resistance to these drugs, which would add relevance to the paper.

-Because it is proposed that CD271 plays an essential role in promoting invasiveness in primary tumors, it is recommended that the authors stratify the TCGA data according to primary versus metastasis when assessing correlations between gene expression and survival.

-It would reinforce the paper if the authors could show that CD271 is expressed in a subset of melanoma clinical samples – at the invading front of primary tumors.

-It is unclear how expression of CD271 is regulated. Understanding the molecular mechanisms that turn CD271 expression ON and OFF would make the paper more appealing. Similarly, it is shown that CD271 confers invasion properties and decrease cell proliferation via distinct mechanisms. However, the understanding of the underlying molecular mechanisms is very poor. How does ICD release is critical for the antiproliferative response and how does the interaction with Trk-A required for loss of adhesion? These questions remain unanswered.

Other comments:

-It is claimed that the orthotopic in vivo model developed by the authors “recapitulates the first steps of melanoma metastasis, namely the break out of cells from the primary tumor”. However, this is not exactly correct. This important step is indeed accompanied by an E-cadherin to N-cadherin switch, however, it is thought that the melanoma cells that make that switch are not the cells that are at the origin of metastasis. These melanoma cells eventually initiate VGP lesions and only once such lesion grow/expand dermally that a subset of melanoma cells acquires metastatic potential through phenotype switching. The authors should rephrase their statement.

-Showing the FACS-sorting profile of Figure 1A would help.

-Figure 1CD is difficult to interpret: Which genes are used for Metacore pathway 1-12 is not clear. Metacore 2 and 6 redundant?

Reviewer #3 (Remarks to the Author):

The Authors studied the roles of low affinity neurotrophin receptor CD271 in phenotype switching of melanoma mainly by analyzing the behavior of human melanoma cells in nude mice. They found that CD271 expression shows good correlation with melanoma invasiveness and inverse correlation with melanoma growth. By modifying the expression level of CD271 in a human melanoma cell line using lentivirus systems, they found the functional link between CD271 and regulation of phenotypic switching of the melanoma cells.

The paper is interesting and is important for understanding of melanoma cell expansion, invasiveness and metastasis. Some of previous studies, however, do not support the Authors' point on the role of CD271 in melanoma invasiveness, as described in the Discussion section of the manuscript. To strengthen the Authors' points related to the controversy, the Authors should address the following points.

1) The authors should provide more precise information about their melanoma cell lines which they used for each figure. Which melanoma cell lines were used for each single experiment should be clear. It seems that they used A375 line mainly, but the line is not necessarily a standard melanoma cell line. More clear description about their experimental system and statistic results for melanoma invasiveness and metastasis are necessary. The role of CD271 in melanoma progression in vivo will be de defined more by using more cell lines.

2) As M050829 line express high level of CD271, siRNA of CD271 in this melanoma cell line will be useful to strengthen their points.

3) B16 melanoma cell line, a syngenic mouse melanoma cell line has been commonly used. Similar lung metastasis can be obtained with the use of B16 melanoma?

- 1) Reviewer #1 did not raise specific comments and stated:

Overall, the work is methodically of very high quality and addresses a clinically highly relevant topic which provides novel insights into the mechanisms of melanoma cell phenotype switching in the process of metastatic progression.

We thank the reviewer for this very positive assessment of our work.

- 2) Reviewer #2, Point 1:

Throughout the paper the authors refer to melanoma “phenotype switching” and study the involvement of CD271 in this process. It is claimed that CD271 can promote the proliferative to invasive switch. However, I am concerned that expression of CD271 is absent from the invasive expression datasets published by others including Hoek and colleagues themselves (45 gene sized INV list) and more recently Verfaillie et al. According to these data CD271 is not a classical invasive gene. Are cells expressing CD271 in a true invasive cell state as previously defined or an intermediary state – or different (de-differentiation) state that has not

been extensively characterized previously?

The issues raised here are very well taken and we agree with the reviewer that it is important to put CD271 and its function in the context of known phenotype switching programs. Indeed, as mentioned by the reviewer, CD271 is not among the classical invasive genes according to the Verfaillie and Hoek programs. In our view it has to be considered that these programs were determined by means of established cell lines that likely represent mainly “late” states of either proliferative or invasive cells at the population level. As we have shown before when analysing properties of CD271-positive melanoma cells (Civenni et al., Cancer Research 2011) and, most importantly, as recently demonstrated by single cell sequencing (Shaffer et al., Nature 2017), CD271 along with other plasticity markers is normally found in a relatively rare cell population of melanoma that we propose is involved in the phenotype switching process. Conceivably, the Verfaillie and Hoek programs do not optimally (if at all) represent cells undergoing phenotype switching. In contrast, CD271 itself is part of the “MITF^{low}/AXL^{high}” program derived from single cell analysis of melanoma tissue *in vivo* (Tirosh et al., Science 2016), consistent with our *in vivo* / *ex vivo* data presented in Figure 1. We now discuss these issues on pp.16/17.

3) Reviewer #2, Point 1 (cont.):

I suggest that the authors establish the transcriptome (and if possible an H3K27ac profile) of melanoma cells expressed CD271 ectopically and compare the gene expression/chromatin signature they obtain with the Hoek and Verfaillie datasets. Or even better, using the TetO system show that the transcriptome of melanoma cells can switch back and forth between the invasive and proliferative cell states by comparing profiles (GSEA) with the ones described by Hoek and Verfaillie.

We highly appreciate this reviewer’s excellent suggestion to establish the global gene expression profiles of melanoma cells before, during and after inducible CD271 expression. Using our TetO system, we performed RNAseq of cells subject to reversible “off-on-off” expression of CD271 (new Figures 7 and 8). As proposed, we then compared the genes dynamically regulated by CD271 with those found in the Hoek, Verfaillie as well as Tirosh (Science 2016; AXL-MITF) datasets. Again, compatible with the above-mentioned explanation regarding CD271 expression, the gene set regulated by CD271 hardly overlapped with the indicated programs, as we now show in Supplementary Figure 6). Likewise, as shown in the new Supplementary Figure 7, melanoma cell xenographs with constant induction of CD271 did not display decreased or absent MITF expression, as one might expect from a tumor comprised of ‘invasive cells’ according to Müller et al (Nat

Commun. 2014). To address this matter in a statistical way, we decided to run GSEA (as suggested by Reviewer #2 as well) for those gene sets that were represented in our CD271 “off-on-off” data sets to at least 5%. This was only the case for the comparison of CD271 “off-on-off” with the Tirosh MITF gene set and the Verfaillie proliferative gene set. However, since the genes overlapping in the two programs were inconsistently up-or downregulated in our transient CD271 (“off-on-off”) program, GSEA did in both cases not result in significant correlation (Supplementary Figure 6). Taken together, as now discussed on pp.16-17, we conclude that short-term transient CD271 might influence processes that are independent of previously reported invasive transcriptional programs; alternatively and favored by us, the CD271-regulated gene set might not be represented in these programs because the latter might primarily reflect established states rather than transient states of cells undergoing phenotype switching. As mentioned above, CD271 itself is part of the Tirosh AXL^{high} program. Conceivably, the program downstream of CD271 might become attenuated with time in AXL^{high} cells; furthermore, not all CD271/AXL^{high} cells might be exposed *in vivo* to NGF, the ligand that most likely stimulates the CD271/Trk-A signalling axis.

4) Reviewer #2, Point 2:

TGFb was used to induce phenotype switching. However, this is not what most people have been using in the field. It is suggested that CD271 expression is analyzed in cells exposed to TNFa, Hypoxia and BRAF/MEK-inhibitors. The latter experiment may allow the authors to study the potential role of CD271 in therapy resistance to these drugs, which would add relevance to the paper.

There are several publications on TGFb and EMT/invasiveness in melanoma, including by our collaborator R. Dummer whose group coined the term “phenotype switching” (Hoek et al, Cancer Res. 2008; Schlegel et al., Exp. Dermatol. 2015). On p.7, when introducing TGFb and CD271, we now also cite a review to clarify this point (Perrot et al., Ann Dermatol. 2013). We agree, however, that other cues can also induce melanoma cell plasticity and, notably, the number of CD271-expressing cells. Apart from TNFa (Landsberg et al., Nature 2012; Hölzel and Tüting, Trends in Immunology 2016), we now also refer to two publications that appeared during the revision of our manuscript (Richard et al., EMBO Mol. Med. 2016; Shaffer et al., Nature 2017), revealing emergence of CD271 expression in cells becoming resistant to BRAF/MEK inhibitors. We have extended the Introduction (p.4) and Discussion (pp. 19/20) accordingly. To functionally address the role of CD271 in therapy resistance would indeed be highly relevant, as pointed out by the reviewer; however, we feel that such quite extensive additional experiments are beyond the scope of the present paper, in which we demonstrate an involvement of CD271 in proliferative-to-invasive phenotype switching.

5) Reviewer #2, Point 3:

Because it is proposed that CD271 plays an essential role in promoting invasiveness in primary tumors, it is recommended that the authors stratify the TCGA data according to primary versus metastasis when assessing correlations between gene expression and survival.

We have now extended our TCGA analysis in the revised Figure 1, by means of analyzing additional invasiveness and differentiation genes with respect to CD271 expression. Moreover, we have performed further TCGA data analysis assessing expression of cholesterol synthesis genes and melanoma patient survival. As outlined below, cholesterol synthesis genes turned out to be regulated by CD271/Trk-A and inhibition of cholesterol synthesis prevented CD271-induced changes in melanoma cell adhesion. Of note, the expression of several of these cholesterol synthesis genes correlates with reduced patient survival, as now shown in the new Figure 7. As suggested by Reviewer #2, we also analyzed TCGA data for CD271 and the neurotrophins NTR3, NGF and BDNF comparing primary tumors versus metastases, which however did not result in significant differences (data not shown). As discussed above, we explain those findings with the fact, that the CD271 high cell population in the tumor is on one hand a small, minor cell population and on the other hand in a transient, dynamic state switching from a proliferative to an invasive state. Therefore other analyses, for example at single cell resolution are in our opinion more informative than TCGA data analysis, which for sure is very powerful and clinically relevant but reflects gene expression averaged over a whole biopsy without cellular and also temporal resolution.

6) Reviewer #2, Point 4:

It would reinforce the paper if the authors could show that CD271 is expressed in a subset of melanoma clinical samples – at the invading front of primary tumors.

Together with our collaborator, R. Dummer (Dermatology, University Hospital Zurich), we tried to assess whether CD271-expressing cells can reliably be associated with the invading front of tumors. However, probably also given the rather low numbers of CD271-expressing cells in human tumors (s. our point 2) above), a clear statement based on a statistically significant correlation is in our opinion difficult to make. Therefore, we chose to transplant melanoma cells in skin substitutes, which allows capturing cells “at the invading front” in a somewhat more controllable manner. Importantly, a link between CD271 protein expression in human samples and metastasis formation has previously been published by our lab

(Civenni et al., 2011; s. Introduction p.4).

7) Reviewer #2, Point 4:

It is unclear how expression of CD271 is regulated. Understanding the molecular mechanisms that turn CD271 expression ON and OFF would make the paper more appealing. Similarly, it is shown that CD271 confers invasion properties and decrease cell proliferation via distinct mechanisms. However, the understanding of the underlying molecular mechanisms is very poor. How does ICD release is critical for the antiproliferative response and how does the interaction with Trk-A required for loss of adhesion? These questions remain unanswered.

We fully agree that understanding how CD271 on – off expression is regulated is an important question. In fact, it is unclear in general how the switch from proliferative to invasive and back to proliferative states is regulated. To the best of our knowledge, our study is actually the first to show that dynamic, reversible expression of a regulatory molecule promotes phenotype switching (from proliferation to invasiveness and to proliferation again), resulting in increased metastasis formation *in vivo*. As addressed in Point 4) above, several cues have been described that induce CD271 expression, but whether these are involved in dynamic CD271 expression or, more generally, in phenotype switching in the 3D context of a tumor *in vivo* remains to be shown.

In the context of our study, we feel that the reviewer's question of how CD271 confers invasion and decreases proliferation is even more interesting. Spurred by the reviewer's suggestion to study the CD271-regulated transcriptome by means of our inducible system (see our Point 3) above), we performed RNAseq-based GO analysis of those genes which were dynamically regulated upon CD271 off-on-off expression (new Figure 7a, b).

Only few genes annotated to 'cell cycle regulation' were controlled by CD271; of note, we found most of these genes to be downregulated upon overexpression of ICD (new Figure 7e and f), consistent with decreased cell cycle progression conferred by CD271-ICD (Figure 6).

Strikingly, the main processes enriched in our GO analysis were associated with cholesterol and lipid biosynthesis. Among cholesterol and fatty acid biosynthesis regulators were genes encoding key enzymes of the cholesterol and fatty acid synthesis pathways, which were all upregulated upon transient CD271 overexpression (Figure 7c). Furthermore, for several of these genes, high expression is associated with worse patient survival according to TCGA (Figure 7d), supporting the clinical relevance of our findings.

Importantly, while ICD did not influence the expression of these cholesterol synthesis genes (Figure 7g), they were all controlled by CD271 in a Trk-A dependent manner (new Figure 8a-

i). Consistent with our data that CD271/Trk-A influences melanoma cell adhesion (Figure 6), addition of cholesterol synthesis inhibitors blocked CD271-mediated cell detachment, demonstrating a functional association between cholesterol synthesis and CD271 function (Figure 8j, k).

These –in our view exciting– novel findings suggest that statins might have to be reconsidered in melanoma treatment in combination with other therapies, as we discuss on pp. 17, 18 and 20.

8) Reviewer #2, Minor Point 1:

It is claimed that the orthotopic in vivo model developed by the authors “recapitulates the first steps of melanoma metastasis, namely the break out of cells from the primary tumor”. However, this is not exactly correct. This important step is indeed accompanied by an E-cadherin to N-cadherin switch, however, it is thought that the melanoma cells that make that switch are not the cells that are at the origin of metastasis. These melanoma cells eventually initiate VGP lesions and only once such lesion grow/expand dermally that a subset of melanoma cells acquires metastatic potential through phenotype switching. The authors should rephrase their statement.

We thank the reviewer for this clarification; we have rephrased the corresponding section accordingly.

9) Reviewer #2, Minor Point 2:

Showing the FACS-sorting profile of Figure 1A would help.

As suggested, we now show, as an example, the FACS profile of one out of two samples and annotated the number of sorted cells for the CD271 high and low populations respectively (Figure 1c).

10) Reviewer #2, Minor Point 3:

Figure 1CD is difficult to interpret: Which genes are used for Metacore pathway 1-12 is not clear. Metacore 2 and 6 redundant?

The pathway terms are defined by MetaCore™ based on the gene set uploaded for analysis. The genes comprised in the given pathways can be found in the gene dataset that will be uploaded to the NCBI's Gene Expression Omnibus after acceptance of the revised manuscript. Also RNA Seq data of the transient CD271 overexpression (Figure 7) will then be available via the European Nucleotide Archive (ENA).

11) Reviewer #3, Point 1:

The authors should provide more precise information about their melanoma cell lines which they used for each figure. Which melanoma cell lines were used for each single experiment should be clear. It seems that they used A375 line mainly, but the line is not necessarily a standard melanoma cell line. More clear description about their experimental system and statistic results for melanoma invasiveness and metastasis are necessary. The role of CD271 in melanoma progression in vivo will be de defined more by using more cell lines.

We thank this reviewer for her/his overall positive and supportive evaluation of our work. Most experiments in the main Figures were performed using the cell line M010817 and not A375. To avoid further confusion in that matter, we now stated in all new figure legends, which cell line(s) were used. In new figures where different cell lines have been used, they are annotated within the figure respectively. As suggested, we now provide more information of the cell lines used (Supplementary Table 2). Importantly, we have also confirmed the key findings of our study using two additional cells lines, as now shown in new Supplementary Figures 3 and 4.

Moreover, we tried to provide a description of our experimental systems by including schemes of the experimental set-up and/or vectors used in the figures.

12) Reviewer #3, Point 2:

As M050829 line express high level of CD271, siRNA of CD271 in this melanoma cell line will be useful to strengthen their points.

Although this is a vey valid suggestion, we were unable to strengthen out points by CD271 loss-of-function studies. As discussed (p.19), others and we observed that CD271 stable knock down fully abolishes growth of melanoma cells for unknown reasons (data not shown).

13) Reviewer #3, Point 3:

B16 melanoma cell line, a syngeneic mouse melanoma cell line has been commonly used. Similar lung metastasis can be obtained with the use of B16 melanoma?

As the focus of our study is on human melanoma cells and also given the difficulty to obtain metastases from subcutaneously (rather than i.v.) injected B16 melanoma cells, we decided to confirm our findings with additional human melnaoma cell lines, as suggested by this reviewer, rather than by using the murine B16 line.

REVIEWERS' COMMENTS:

Reviewer #2 (Remarks to the Author):

The authors have addressed all my concerns and criticisms adequately.

Reviewer #3 (Remarks to the Author):

The Authors improved their manuscript by responding to Reviewers' comments. Overall the manuscript has been greatly improved and is now appropriate for publication.